# Tribology Performance of Surface Texturing Plunger

**DOI:** 10.3390/biomimetics4030054

**Published:** 2019-08-05

**Authors:** Songbo Wei, Hongfei Shang, Chenglong Liao, Junyuan Huang, Bairu Shi

**Affiliations:** 1PetroChina Research Institute of Petroleum Exploration and Development, Beijing 100083, China; 2State Key Laboratory of Tribology, Tsinghua University, Beijing 100084, China

**Keywords:** tribology performance, non-smooth surface, surface texturing, plunger, pumping unit

## Abstract

Plunger pumps are widely used in oil pumping units around the world. The water content of the wellbore is increasing along with the development progress, so the lubricating capacity of the well fluids between the plunger and barrel is decreasing correspondingly. Commonly, the substrate material of the plunger and barrel are stainless steel, and the plunger surface is usually covered with nickel-based coating. Therefore, the performance of the plunger and barrel has been affected due to poor lubrication and eccentric wear. Non-smooth surfaces have been proven to improve the tribology performance in many cases. A surface texturing plunger covered with specific dimples has been prepared by using laser surface texturing technology. The morphology of the surface texturing plunger was characterized and analyzed. The tribology performance of surface texturing plunger samples was tested using standard friction and wear test machines with oil and water lubrication, respectively. The results indicated that surface texturing could effectively reduce the coefficient of friction, and the wear resistance of the surface textured samples has been improved to some extent.

## 1. Introduction

There are around more than one million mechanical recovery wells, 90% of which are plunger pump wells. There are many advantages to the plunger pump, such as simple mechanical structure, convenient operation, and longer service life in comparison with other lifting styles. Currently, most oilfields have entered the middle and late stages of development, and many problems are induced by sand production, high water cut, and so on, which results in wear loss, leakage, corrosion, and other failures of pumps [1]. Therefore, the inspection cycle of pumps has become shorter and the overall economic efficiency of oilfield development is seriously affected.

The deterioration of fluid supply capacity of wells and the increase of pump setting depth lead to partial abrasion between the cylinder and the plunger, seriously reducing the service life and detection period of the pump. When oilfields enter the middle and late stages of development, the water content of the production liquids becomes high, and the lubricating capacity of the well liquids decreases [2]. The friction coefficients between the pump barrel and plunger increase, and the wear rates of the pump barrel and plunger accelerate. Finally, the leakage of production liquids increases, which results in a decrease in pump efficiency. The sand problem becomes prominent for many oil wells currently. Sand content becomes seven to eight times higher than that of the early stages. High sand content will cause sand plugs, abrasion, deformation, and other problems.

At present, there are many methods to improve the wear resistance of the barrel/plunger friction pairs. Nickel or chromium based coatings with high hardness and wear resistance are usually employed on the plunger. Some plungers have been machined into annular grooves. The annular grooves could store sand in the gap between the plunger and barrel, thus reducing the wear caused by the sand. The annular grooves also could store oil to lubricate the barrel and plunger and also help to dissipate heat caused by friction.

In recent years, non-smooth surfaces have been intensively studied in reducing friction and improving anti-wear properties in mechanical components [3,4,5,6]. Laser surface texturing (LST) technology is widely used to fabricate specific non-smooth surfaces, and the effect of some surface textures on reducing friction and improving anti-wear properties has been theoretically predicted and experimentally verified [7,8,9]. Etsion’s studies indicated that the LST enhanced hydrodynamic and hydrostatic lubrication of mechanical seals analytically and experimentally, and there was a significant improvement in the load capacity, anti-wear and friction coefficient, and a substantial reduction in wear and surface damage in comparison with untextured surfaces [10,11]. Li’s study also indicated that the textured titanium surface exhibited better anti-friction and anti-wear performance under the small particle wearing condition compared with that under the large particle [12].

The design of surface texture is also inspired by the animals in the wild. The non-smooth surface textures of elytra have shown superior performance of friction reduction, and the results of further tribological experiments show that the circle texture had the lowest friction coefficient [13]. The friction and wear properties of steels with and without bionic non-smooth surfaces under dry/lubricant wear conditions were evaluated, and the bionic surfaces of testing samples had better wear resistance and provided stable friction coefficient to some extent [14].

There was rare report on the laser surface textures used in plunger pumps. In this work, non-smooth surface of plunger with specific dimples was prepared by using laser surface texturing technology. The experimental investigation of the effect of LST on the friction between barrel and plunger was studied. The morphology of the surface texturing plunger was characterized and analyzed before and after wear tests. The tribological performance of surface texturing plunger samples was tested by using standard friction and wear test machine with the oil and water lubricating, respectively.

## 2. Materials and Methods

A plunger with nickel-based coating of about 120 μm in thickness was used as a substrate. Round dimples with a diameter of about 100 μm and a depth of about 10 μm were fabricated on the plunger surface using laser surface texturing technology. In order to facilitate the wear tests, the dimple surface textures were also fabricated on the surface of the stainless steel sample. The output power of the laser source was 10 W with a center wavelength λ of 1064 nm. Four kinds of texturing samples with the area densities of 50, 55, 60, and 65% were prepared.

The tribological performance of the samples with different surface textures was characterized by the ball-disk test by using the SRV-4 high temperature friction and wear tester. The sample of upper friction pair was a GCr15 steel ball with a diameter of 10.2 mm. The sample of lower friction pair had a diameter of 24 mm and a thickness of 7.88 mm and had four texture areal densities: 0, 50, 55, 60, and 65%, respectively. As a comparison, a sample with a non-textured surface was also tested.

In the friction and wear tests, the upper sample had a reciprocating frequency of 20 Hz, a stroke of 1 mm, and a load of 5 N. The experiment was carried out for 30 min at room temperature, and the lubricating media were base oil (PAO6) and deionized water, respectively. The micro-morphologies and structures of the samples were characterized by a white light interferometer. 

## 3. Results and Discussion

Figure 1a shows a partial area of the plunger covered with dimple laser surface textures, and Figure 1b shows an optical micrograph of the dimples. It can be seen that all dimples were consistent in appearance and uniformly distributed on the surface. During the laser processing, because irradiation of the central region the materials were melted and vaporized, and there were some melted materials accumulated around the dimple margin, as shown in Figure 1b.

The curve of the friction coefficients with the base oil lubricating as function of time is shown in Figure 2. The friction coefficients of the sample with non-textured surfaces were above 0.4, and most of time the friction coefficients were approximately 0.3. For the samples with surface textures, the friction coefficients were about 0.18, and there was no obvious difference among the four textured samples. The difference in friction coefficients between samples with different texture area ratios is not obvious. It can be seen that the friction coefficients of textured samples were lower than that of the sample with a non-textured surface. The results indicate that surface textures with the oil lubricating could reduce the friction between friction pairs.

Figure 3 shows the three-dimensional topography of the wear marks after the test with base oil lubrication. The sample with a non-textured surface showed a more obvious wear scar, and the depth and width of the wear scar were about 5 and 200 μm, respectively. For the four textured samples, the wear scar on each sample was quite slight, and the dimples were not wiped after the tests. The degree of wear on the surface textured samples was less than that of the non-textured sample, which means that the surface textures improved the wear resistance of the substrate materials with the oil lubricating.

Figure 4 shows the curve of friction coefficients with the water lubricating as a function of time. It can be seen that the non-textured sample had a friction coefficient of above 0.5, which was higher than that of the surface textured samples. For the sample with a textured surface with an areal density of 65%, the friction coefficients increased from 0.3 to 0.5, which was higher than that of the other textured samples. The friction coefficients of the samples with textured surface with an areal density of 50, 55, and 60% were approximate and were around 0.3. These results indicated that surface textures with the water lubricating also could reduce the friction between frictional pairs, although the lubricating effect of water was weaker than that of the oil.

Figure 5 shows the three-dimensional topography of the wear marks after the test with the water lubricating. The sample with a non-textured surface showed a more obvious wear scar, and the depth and width of the wear scar were about 5 and 400 μm, respectively. The wear scars on the four textured samples were quite slight, and the dimples were still on the surface after the tests. The wear degree of the surface textured samples was less than that of the non-textured sample, which means that the surface textures also improved the wear resistance of the substrate materials with water lubrication. Under the condition of fluid lubrication, the hydrodynamic lubrication effect caused by surface texture is one of the important reasons for friction reduction. From the mechanism of the hydrodynamic lubrication effect, it can be seen that two parallel smooth surfaces with relative motion cannot form a pressure oil film because there is no convergence clearance, so it cannot produce hydrodynamic bearing capacity [15]. The friction pairs are directly in contact with each other, and there is not enough oil lubrication at interfaces between the friction pairs, which causes high friction and severe wear, as shown in Figure 2 and Figure 3a. 

Surface texture can provide regular convergence gaps for two parallel surfaces. As shown in Figure 6, when the lubricant enters the area of the dimple surface textures due to the relative motion with a velocity *U* between friction pairs, a positive pressure *P* of the lubrication film will occur in the convergence gap and will decrease in the divergence gap, which will produce an asymmetric pressure distribution in each area of dimple, so that the lubrication film has a certain degree of pressure [16,17]. Each dimple corresponds to one micro-bearing of hydrodynamic lubrication, which can produce an additional hydrodynamic lubrication effect and improve the overall lubrication performance of the surface. This is why the friction coefficients of the surface textured samples were lower than that of the smooth samples, as seen in Figure 2 and Figure 4. Correspondingly, the wear resistance was improved by the surface textures, as shown in Figure 3 and Figure 5.

The base oil showed better friction reduction performance than the water, because the base oil could form a thicker oil film on the surface of the friction pairs. Additionally, the viscosity of base oil is higher than that of water, so the base oil has a stronger bearing capacity for the friction pairs. The surface textures show excellent friction reduction performance with base oil lubrication.

## 4. Conclusions

In conclusion, dimple surface textures were fabricated on the plunger surface. The tribology performance of surface texturing plunger samples was tested with oil and water lubrication. The friction coefficients of textured samples were all lower than that of the sample with a non-textured surface, with whether oil or water lubrication was used. With oil lubrication the friction coefficients of the textured samples were about 0.18, while the friction coefficients of the non-textured samples reached 0.3. The wear scars on the textured samples were quite slight, and the dimples were still on the surface after the tests. The surface textures obviously improved the wear resistance of the substrate materials. The hydrodynamic lubrication effect caused by surface texture is the main reason for friction reduction.

## Figures and Tables

**Figure 1 biomimetics-04-00054-f001:**
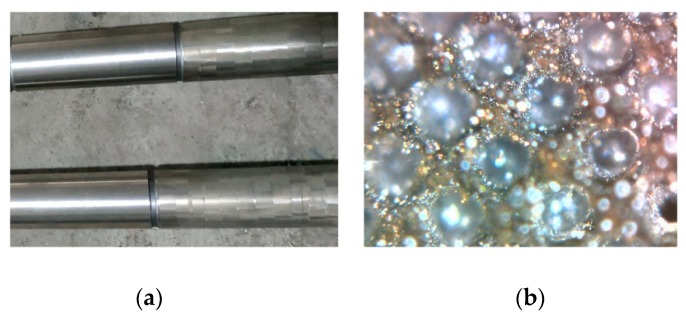
(**a**) Plunger surface on partial area covered with dimple laser surface textures, (**b**) the optical micrograph of the dimples.

**Figure 2 biomimetics-04-00054-f002:**
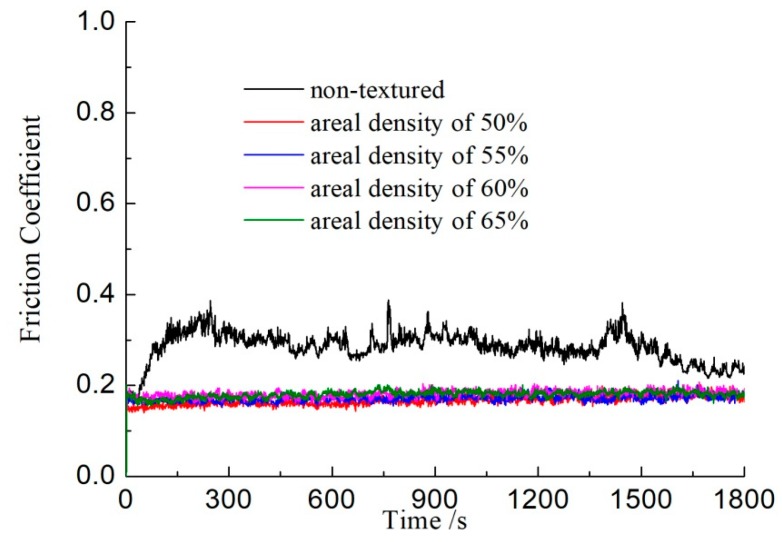
The curve of friction coefficients with the base oil lubricating as a function of time.

**Figure 3 biomimetics-04-00054-f003:**
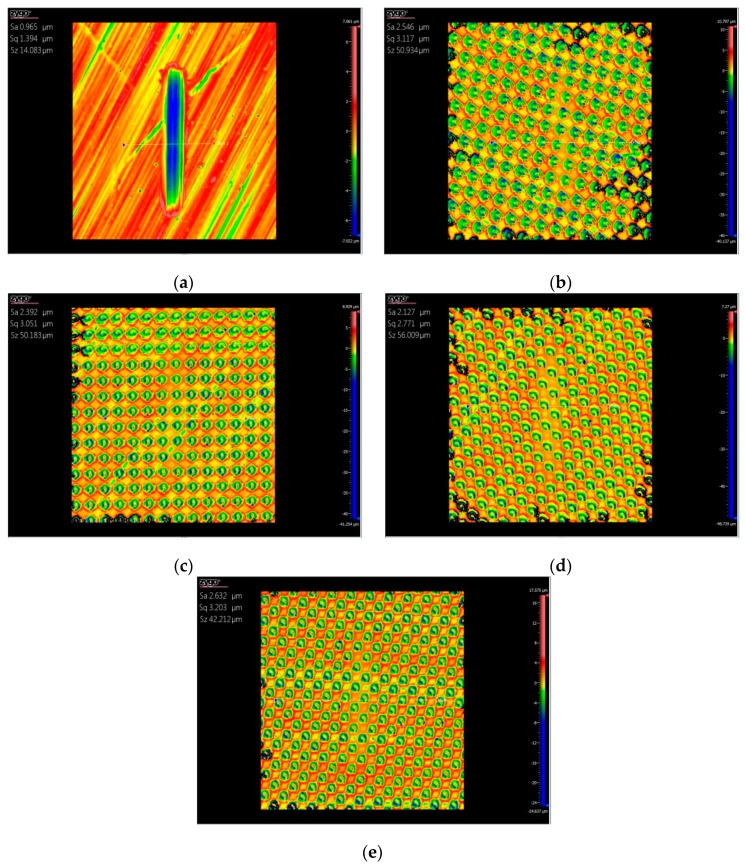
Three-dimensional topographies of the sample after wear tests with the base oil lubricating, (**a**) non-textured sample, surface textured samples with an areal density of (**b**) 50, (**c**) 55, (**d**) 60, and (**e**) 65%.

**Figure 4 biomimetics-04-00054-f004:**
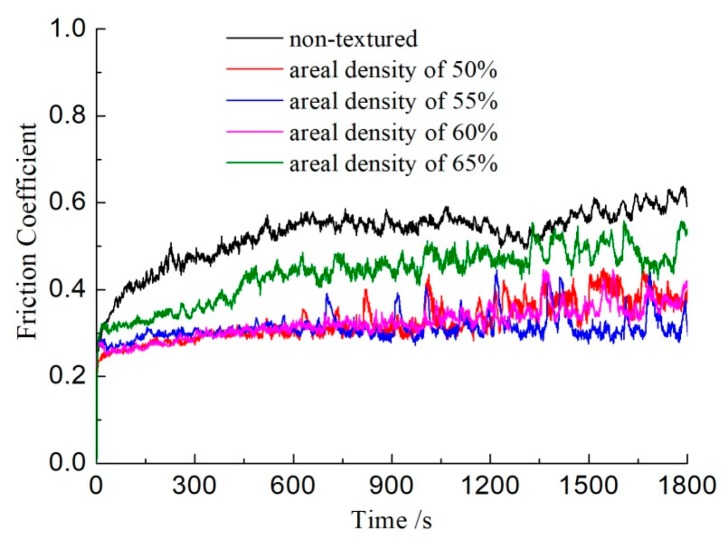
The curve of friction coefficients with the water lubricating as a function of time.

**Figure 5 biomimetics-04-00054-f005:**
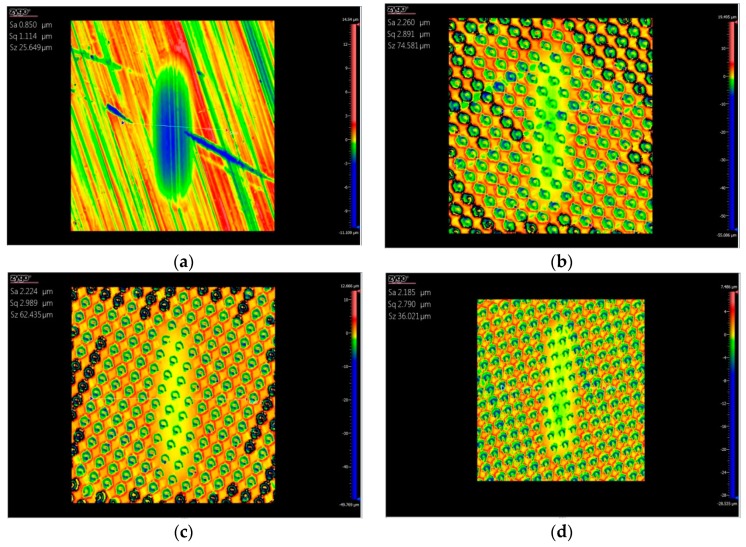
Three-dimensional topographies of the samples after wear tests with the water lubricating: (**a**) Non-textured sample, surface textured samples with an areal density of (**b**) 50, (**c**) 55, (**d**) 60, and (**e**) 65%.

**Figure 6 biomimetics-04-00054-f006:**
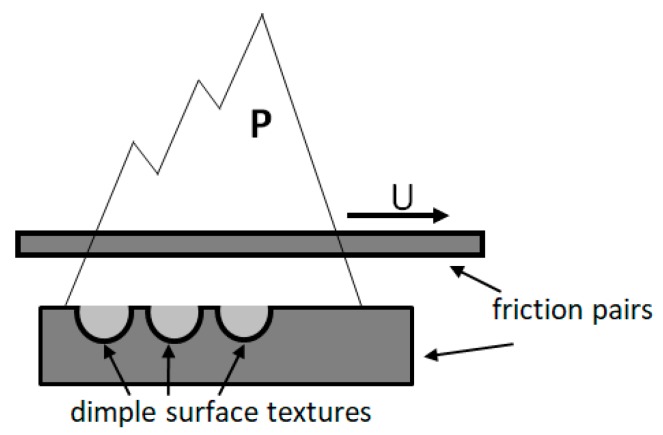
The schematic illustration of the hydrodynamic lubrication effect for the dimple surface textures.

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
