# Peer review of "Tribology Performance of Surface Texturing Plunger"

_biomimetics, 2019, doi:10.3390/biomimetics4030054_

Round 1

Reviewer 1 Report

This work provides a collection of experimental analysis concerning the influence of using surface texturing technology on reducing friction and wear between plunger and barrel in plunger pumps. Surface texturing in form of dimples was prepared by laser and tribological performances were evaluated on a standard friction test machine under lubricated contact conditions. As reported in the paper, surface texturing reduces friction and increases wear resistance to some extent. The following comments/questions must be addressed before publication in the journal biomimetics.

English needs revision

The used materials (plunger and barrel) should be     indicated in the abstract.

Introduction is VERY POOR, only 9 references in     the introduction! Should be improved and particularly concerning surface     texturing (in the literature there are many recent publications dealing     with LST).

In the introduction section, the prior work     survey is weakly presented. Authors should illustrate the major     conclusions of the prior works that they cited.

Introduction: line 49 the reference should be [3-4]     not 34. Same remark in line 55 the reference should be [8-9] not 89!!

The originality of the paper should be more clarified     in the last paragraph in the introduction, what is different comparing to     others works reported in the literature.

Section Materials and Methods should be improved.     What is the depth and the shape pf the dimples? Why did authors choose 50%,     55%, 60%, and 65%, was this based on a theoretical model or previous     results (if yes this should be explained). The authors should explain the     process of samples preparation, surface characterizations….

Why did authors test the tribological proprieties     on a reciprocal testing machine with ball-on flat contact knowing that in     real pumps the plunger-barrel contact is a conformal?

As mentioned in the introduction, one of the     major problems is the presence of sand in the liquid, did authors try to     add particles in the lubricants to simulate this effect?

Schematic description of the used tribometer as     well as test procedure are missing! How tests were conducted? Did authors     perform any running-in prior to test? could authors show typical variation     of friction during a single cycle (one loop)?

Section results and discussion: Could authors     explain how did they obtain the results presented in figure 2. Is this this     average friction coefficient? Indeed, in reciprocal motion the value of     the friction force and relative velocity change with respect to displacement     and thus the friction coefficient changes as well.

Lines 91-92: authors mentioned “The results     indicated that surface textures with oil lubricating could reduce the     friction between friction pairs”. This is an obvious result! They should     explain why does this happen? what is the physical (tribological) mechanism     behind? Same remark concerning the reduction of wear obtained with     textured surfaces, what are the wear mechanisms involved in the two cases     (textured and non-textured samples).

Could authors explain the difference performances     between oil and water in term of lubrication capacity.

Figure 4: it seems that sample 50 55 60 are close     to each other’s in term of performance. It seems that 50 is enough to achieve     the maximum friction reduction. Did authors try another sample with less     texturing (aspect ratio)? could they discuss this point?

Line 124: the references should be [11 - 12] not     1112

Explanation in the paragraph between lines 121 and     129 should be with schematic illustration, this can be helpful for general     readers.

Author Response

Response to Reviewer 2 Comments

Point 1: The used materials (plunger and barrel) should be indicated in the abstract.

Response 1: we added the materials in the abstract in the revised manuscript.

Point 1: Introduction is VERY POOR, only 9 references in     the introduction! Should be improved and particularly concerning surface     texturing (in the literature there are many recent publications dealing  with LST).In the introduction section, the prior work  survey is weakly presented. Authors should illustrate the major  conclusions of the prior works that they cited.

Response 2: we had revised the Introduction section in the revised manuscript.

Point 1: Introduction: line 49 the reference should be [3-4]     not 34. Same remark in line 55 the reference should be [8-9] not 89!!

Response 3: we have revised these problems in the revised manuscript.

Point 1:The originality of the paper should be more clarified  in the last paragraph in the introduction, what is different comparing to  others works reported in the literature.

Response 4: We have revised the last paragraph in the revised manuscript.

Point 1:Section Materials and Methods should be improved.     What is the depth and the shape pf the dimples? Why did authors choose 50%,     55%, 60%, and 65%, was this based on a theoretical model or previous results (if yes this should be explained). The authors should explain the     process of samples preparation, surface characterizations….

Response 5: we have revised these problems in the revised manuscript. The round dimples with the diameter of about 100 μm and the depth of about 10 μm were fabricated on the plunger surface by using laser surface texturing technology. We have done many experiments about the areal ratio, in this work we choose typical parameters like 50%, 55%, 60%, and 65% to study. The process of sample preparation and characterization are quite conventional, so we only supplied the main parameters but not the details. The output power of the laser source was 10 W and the center wavelength λ of 1064 nm. The micro- morphologies and structures of the samples were characterized by white light interferometer.

Point 1:Why did authors test the tribological proprieties     on a reciprocal testing machine with ball-on flat contact knowing that in     real pumps the plunger-barrel contact is a conformal?

Response 6: In this study we employ standard test machine to do the tests, because  this method was easy to carry out, and could simulate plunger motion to some degree, and could test the tribological effect of the laser surface textures.

Point 1:As mentioned in the introduction, one of the     major problems is the presence of sand in the liquid, did authors try to add particles in the lubricants to simulate this effect?

Response 7: In this work we didn’t add particles during the tests. To add particles in the lubricants would make the experiments more complicated. We did the tests in the conventional and ideal conditions. As to the particles participation conditions, we will do this work in the future.

Point 1:Schematic description of the used tribometer as  well as test procedure are missing! How tests were conducted? Did authors  perform any running-in prior to test? could authors show typical variation of friction during a single cycle (one loop)?

Response 8: The tribological performance of the samples with different surface textures was characterized by the ball-disk test by using the SRV-4 high temperature friction and wear tester. The sample of upper friction pair was a GCr15 steel ball with a diameter of 10.2 mm. The sample of lower friction pair had a diameter of 24 mm and a thickness of 7.88 mm. We recorded the data from the beginning of the friction tests, and the curves of friction coefficients as a function of time were shown in Figure 2 and 4.

Point 1:Section results and discussion: Could authors explain how did they obtain the results presented in figure 2. Is this average friction coefficient? Indeed, in reciprocal motion the value of     the friction force and relative velocity change with respect to displacement     and thus the friction coefficient changes as well.

Response 9: The SRV-4 standard machine automatically recorded the data and friction coefficient as function of time during the tests. The value of  the friction force and relative velocity transiently change with respect to displacement, so it is always the average friction coefficient we could get. The friction coefficient given from the SRV-4 machine could reflect the difference between different samples to some degree. So the results of the friction tests were convincing.

Point 1:Lines 91-92: authors mentioned “The results     indicated that surface textures with oil lubricating could reduce the     friction between friction pairs”. This is an obvious result! They should     explain why does this happen? what is the physical (tribological) mechanism     behind? Same remark concerning the reduction of wear obtained with     textured surfaces, what are the wear mechanisms involved in the two cases     (textured and non-textured samples).

Response 10: we have explained the mechanism from line 121 to 141 in the revised manuscript.

Point 1: Could authors explain the difference performances     between oil and water in term of lubrication capacity.

Response 11: We have added the explanation of difference performances between oil and water in term of lubrication capacity in the revised manuscript. The oil has better performances of lubrication capacity.

Point 1:Figure 4: it seems that sample 50 55 60 are close     to each other’s in term of performance. It seems that 50 is enough to achieve     the maximum friction reduction. Did authors try another sample with less     texturing (aspect ratio)? could they discuss this point?

Response 12: In the first 400 s, the samples of areal density 50% and 60%  exhibited better friction reduction. After 700 s, the sample of areal density 55% achieved the maximum friction reduction, so we didn’t try other areal densities in this work. For further research, we will do more study about other areal densities.

Point 1:Line 124: the references should be [11 - 12] not     1112

Response 13: we have revised this problem in the revised manuscript.

Point 1:Explanation in the paragraph between lines 121 and     129 should be with schematic illustration, this can be helpful for general     readers.

Response 14: we have added the schematic illustration in the revised manuscript.

Reviewer 2 Report

Review report of article “Tribology performance of surface texturing plunger”.

 In the current manuscript, authors prepared the surface textured plunger with specific dimples using laser surface texturing. Later the tribology of surface textured plunger samples is analyzed using standard friction and wear test with oil and water lubrication. The reviewer suggests some major corrections before publication.

1.    Why the authors didn’t consider the variation of load and frequency for the work.

2.    Authors should do SEM analysis for a better explanation of the results.

3.    The reviewer didn’t find the discussion for specific wear calculations.

4.    XRD should be performed to check whether the phase composition changed or not?

5.    References should be properly inserted with the use of [ ].

6.    Reviewer suggests the authors to reproduce the experimental data in software like origin or in excel. Because it’s difficult to see in the pictures generated by the software and the quality is too bad.

7.    Please refer some relevant papers: “Tribological behavior of textured titanium under abrasive wear” DOI: 10.1080/02670844.2018.1512233

8.    Inline 114-120 line spacing is different.

9.    Insert the pictures with text, rather than putting all the pictures at the end before the conclusion.

Author Response

Response to Reviewer 1 Comments

Point 1.   Why the authors didn’t consider the variation of load and frequency for the work.

Response 1: We are carrying out the experiments with about other factors such as load, frequency, lubricating medium and so on. The experimental results and conclusions would be published in the future papers.

Point 2.   Authors should do SEM analysis for a better explanation of the results.

Response 2: The SEM analysis could show the detailed structures of the samples. In this work, the three-dimensional micro-morphologies and structures of the samples were clearly characterized by using white light interferometer. We also have done the SEM analysis, but the SEM micro-morphologies of the dimples were not clear enough due to the small depth-diameter ratio of the dimple and quite light wear scars.  The authors think the three-dimensional micro-morphologies are enough to explain the experimental results.

Point 3.   The reviewer didn’t find the discussion for specific wear calculations.

Response 3: In this work, the wear scars on the textured samples were quite slight, and it is hard to calculate the abrasion lose, so there was no specific wear calculations. For the untextured samples, the depth and width of the wear scar were about 5 μm and 200 μm for the oil lubricating, and the depth and width of the wear scar were about 5 μm and 400 μm for the water lubricating. The depth and width of the wear scar could be measured from the outline of the three-dimensional micro-morphologies.

Point 4.  XRD should be performed to check whether the phase composition changed or not?

Response 4: In this paper the authors didn’t employ XRD to check the change of the phase composition. The main purpose of this paper focused on studying the friction and wear of the textured surface with specific parameters, and to use the hydrodynamic lubrication effect of the textured surface to improve the overall lubrication performance of the surface. The phase composition might have changed in this work, in the future work we will do more work to study these details.

Point 5.  References should be properly inserted with the use of [ ].

Response 5: We use the template of the journal biomimetics.

Point 6.  Reviewer suggests the authors to reproduce the experimental data in software like origin or in excel. Because it’s difficult to see in the pictures generated by the software and the quality is too bad.

Response 6: We have revised some bad quality pictures.

Point 7.  Please refer some relevant papers: “Tribological behavior of textured titanium under abrasive wear” DOI: 10.1080/02670844.2018.1512233

Response 7: We have referred this paper in the revised manuscript.

Point 8.  Inline 114-120 line spacing is different.

Response 8: We had revised this problem in the revised manuscript.

Point 9.  Insert the pictures with text, rather than putting all the pictures at the end before the conclusion.

Response 9: We use the template of the journal biomimetics.

Round 2

Reviewer 1 Report

Introduction needs to be improved 

add more references

Author Response

Thanks for the good suggestions of the reviewer. We had added several new references and revised the Introduction in the revised manuscript.

Reviewer 2 Report

Based on authors comments, I am ok with the manuscript.

Author Response

Thanks the reviewer very much for the good suggestions.